# Retrospective Analysis to Optimize the Detection of *MET* Exon 14 Skipping Mutations in Non-Small Cell Lung Cancer

**DOI:** 10.3390/diagnostics14111110

**Published:** 2024-05-27

**Authors:** Jang-Jih Lu, Shu-Hui Tsai, Lee-Chung Lin, Tzong-Shi Chiueh

**Affiliations:** 1Department of Laboratory Medicine, Chang Gung Memorial Hospital, Lin-Kou, 5 Fu-Shing St. Kweishan, Taoyuan 333, Taiwan; 2Department of Medicine, College of Medicine, Chang Gung University, 259 Wen-Hwa 1st Rd. Kweishan, Taoyuan 333, Taiwan

**Keywords:** *MET*ex14, oncomine focus assay, reverse transcription PCR

## Abstract

Our study optimized *MET*ex14 skipping mutation detection by analyzing 223 Oncomine^™^ Focus Assay-positive cases using Pan Lung Cancer PCR Panel and reverse transcription (RT)-PCR. Among the 11 *MET*ex14 skipping mutation-positive cases (average read counts: 1390), 2 with Oncomine^™^ Focus Assay read counts of 2540 and 10,177 were positive on all platforms. Those with Oncomine^™^ Focus Assay read counts ranging from 179 to 612 tested negative elsewhere. Specimens with low ratios (average ratio: 0.12% for nine cases) may yield false-positive results. Our results suggested that monitoring read counts and ratios and validating the results with RT-PCR are crucial to prevent false positives.

## 1. Introduction

The mesenchymal–epithelial transition factor (*MET*) gene is a proto-oncogene containing 21 exons, located on chromosome 7 in the 7q21 to 7q31 region, and encodes a receptor tyrosine kinase. Through controlling the activation of several signaling pathways, such as RAS/MAPK, Rac/Rho, and PI3K/AKT, it is involved in the regulation of various cellular processes including cell growth, survival, and migration. However, *MET* is aberrantly overexpressed in specific types of cancers [1], and 2–3% of lung adenocarcinomas and approximately 1% of lung squamous cell carcinomas exhibit aberrant *MET* overexpression [2]. Previous studies have shown that the exon 14 of the *MET* encodes a regulatory domain that prevents *MET* overexpression. Point mutations within exon 14, such as Y1003X or D1010X, which cause the *MET* overexpression and lead to disrupt normal signaling pathways, promote uncontrolled cell growth and proliferation, contributing to the development and progression of lung cancer, particularly non-small cell lung cancer (NSCLC) [2]. 

It has been found that patients with the *MET* exon 14 (*MET*ex14) skipping alteration have a significantly poor prognosis, making the identification of this mutation crucial for an accurate diagnosis and personalized treatment strategies [3,4]. Therefore, efficient biomarker detection methods are necessary for this mutation identification. The next-generation sequencing (NGS) technology which has arisen and been developed over last two decades has become an useful tool adapted in many biological applications, especially for medical diagnostics. Recently, accompanied by the tyrosine kinase inhibitors developed and applied in clinical treatment, companion diagnostics (CDx) kits were acquired to monitor and evaluate their treatment efficacy. Based on this purpose, several CDx have been developed to assess the treatment efficacy of these anticancer medication, such as FoundationOne^™^ [5,6] (Roche, Basel, Switzerland), ArcherMET^™^ [7,8] (Invitae, San Francisco, CA, USA), or Oncomine^™^ Focus Assay (Thermo Fisher Scientific, Waltham, MA, USA) [9,10]. Each CDx were developed by different strategies and are run on various NGS platforms, such as FoundationOne^™^ is designed to detect 324 potential cancer-related genes through the Illumina^®^ platform. Moreover, the Oncomine^™^ Focus Assay is designed to monitor 52 key solid tumor genes through the Ion Torrent^®^ platform. There are key differences between these two platforms; Illumina offers higher accuracy with a lower error rate, while Ion Torrent may struggle with homopolymers, yet it provides longer reads, which can be advantageous for certain applications like de novo sequencing or transcriptome assembly. Currently, only FoundationOne^™^ CDx has been approved by the Food and Drug Administration of the USA [11] and ArcherMET^™^ approved in Japan; however, none of them have been approved in Taiwan. Therefore, most *MET*ex14 detection methods in Taiwan are laboratory-developed tests (LDTs). Recently, various detection methods for *MET*ex14 were compared, including the commercial CDx kits and LDTs, and it was observed that all detection methods exhibited a high frequency of false-positive results (30.8%) [7]. It is important to find out and eliminate those false-positive cases to avoid unnecessary medication. Since our hospital uses similar strategies for *MET*ex14 detection, it is possible that false-positive results have been reported for our patients, and it is crucial to assess our detection methods for *MET*ex14. This study evaluated all positive cases tested using OFA and compared the results with those of the Pan Lung Cancer PCR Panel (AmoyDx^®^, Xiamen, China) [8] and reverse transcription (RT)-PCR to optimize the routine testing for *MET*ex14.

## 2. Materials and Methods

This retrospective cohort study analyzed previously stored anonymized paraffin-embedded tissues from NSCLC positive cases at the Chang Gung Memorial Hospital, spanning 2019 to 2021. Tissue blocks that met any of the following criteria were excluded: (1) tumor cellularity (tumor content) below 30%, (2) tumor area less than 125 mm^2^, (3) fewer than five blocks or blocks without a 10 micrometer thickness. A total of 223 NSCLC cases at the Chang Gung Memorial Hospital from 2019 to 2021 were assessed using the Oncomine^™^ Focus Assay. The paraffin-embedded tissues served as the material for RNA extraction, which was then analyzed using the Oncomine™ Focus Assay. All positive Oncomine^™^ Focus Assay cases containing the *MET*ex14 were further assessed using the Pan Lung Cancer PCR Panel and LDT RT-PCR. Next-generation sequencing (NGS) data from these three platforms were compared to validate the presence of *MET*ex14. The Oncomine^™^ Focus Assay and Pan Lung Cancer PCR Panel analyses were performed following the manufacturer’s instructions. The DNA and RNA were extracted from paraffin-embedded tissues and used to generate libraries for NGS. After assessment using the Oncomine^™^ Focus Assay, the Ion PGM Hi-Q Sequencing Kit was used along with the Ion PGM system (Thermo Fisher Scientific), and the sequencing results were analyzed using the Ion Reporter software version 5.10. For the Pan Lung Cancer PCR Panel analysis, the library was prepared using the QuantStudio 12 K Flex Real-Time PCR System (Thermo Fisher Scientific). The LDT RT-PCR analysis procedure was developed by our team; the total RNA was extracted using the RNeasy OFFPE kit (Qiagen, Hilden, Germany), and then RT-PCR was performed. Specific primer sets for the RT-PCR were *MET*ex14 FWD: 5′-TTGGGTTITTCCTGTGGCTG-3′ and *MET*ex14 REV: 5′-GGATACTGCACTTGICGGCA-3′. The one-step RT-PCR conditions were as follows: 50 °C, 15 min for cDNA synthesis; 95 °C, 2 min for enzyme inactivation; 45 cycles at 95 °C, 20 s for denaturation; 60 °C, 30 s for annealing; 72 °C, 60 s for extension; and finally extension at 72 °C for 5 min.

## 3. Results

### 3.1. Validation of Oncomine^™^ Focus Assay Results

Common thresholds for a reliable fusion call are fusion read counts larger than 120 and total mapping fusion reads of more than 20,000. Eleven Oncomine^™^ Focus Assay-positive cases had an average read count of 1390 (154–10,177), all of which fit the protocol criteria. We also used the Integrative Genomics Viewer (IGV, version 5.01; https://software.broadinstitute.org/software/igv/, 25 March 2020) to examine the existence of *MET*ex14 skipping mutations after alignment with the human genome (Figure 1). All Oncomine^™^ Focus Assay-positive cases showed similar results when the quality of the read alignment and variant calls was visually inspected.

### 3.2. Validation of the Pan Lung Cancer PCR Panel Results

Results from the Pan Lung Cancer PCR Panel assessments were analyzed according to the manufacturer’s criteria of a cycle threshold (Ct) value < 28, which is considered positive. Two samples (FOCUS-018 and FOCUS-182) that fit the criteria were considered positive for *MET*ex14 skipping mutations (Figure 2). In addition to these two samples, the other samples with Ct values greater than 28 were considered negative for *MET*ex14 skipping mutations.

### 3.3. Validation of RT-PCR Results

Among the 11 Oncomine^™^ Focus Assay-positive cases, only FOCUS-018 and FOCUS-182 showed RT-PCR product sizes identical to those of the *MET*ex14 positive control. FOCUS-012 and FOCUS-157 exhibiting PCR product sizes identical to those of *MET* wild-type specimens were considered negative for *MET*ex14 cases (Figure 3).

### 3.4. Comparative Analysis of Three Platforms Results

We retrospectively analyzed 11 Oncomine^™^ Focus Assay-positive cases that were validated using the Pan Lung Cancer PCR Panel and LDT RT-PCR analyses, and our analysis data showed that a concordance rate of 2 positive cases in all these platforms was 18.2%, and the respective read counts for these two positive samples (FOCUS-018 and FOCUS-182) were 10,177 and 2540 (Table 1). Additionally, the remaining nine samples were also positive using the Oncomine^™^ Focus Assay, and their read counts ranged from 179 to 612 and fit the criteria. However, they tested negative on the other two platforms, indicating discrepancies in the results for samples with lower read counts across different platforms. Nevertheless, two of these nine samples showed a slight increase in amplification curve visibility in the Pan Lung Cancer PCR Panel assessments, with Ct values of 29.8 and 32.2. Slightly above the threshold, this suggests the possibility of minor variations and can be considered as samples of concern. Further analysis by dividing the read count by the total fusion reads showed a consistent result of 4.65% and 1.28% for the two positive samples (FOCUS-018 and FOCUS-182). The average ratio of the remaining nine samples was 0.12%, indicating a significant numerical difference.

## 4. Discussion

The NGS-based companion diagnostics kits have provided a precise way to validate a broad range of target genes within a limited number of specimens, especially since most of them were designed as multiplex gene panels to efficiently monitor multiple genes at a time. This provides valuable therapeutic information during the administration of anticancer medication. Since different companion diagnostics kits were developed and run through various the NGS platform, they caused certain cost price variations which became the other evaluation point for clinical usage [12]. Therefore, much lower price companion diagnostics kits like Pan Lung Cancer PCR Panel are chosen and widely used in many hospitals. However, diagnostic accuracy is still required throughout the anticancer medication process, which led us to consider the target specificity among the different companion diagnostics kits.

In this retrospective study, the occurrence rate of *MET*ex14 skipping in NSCLC was statistically determined to be 4%, similar to what was previously reported [13,14]. However, the potential influence of variant occurrence rates across different ethnicities must be considered, increasing the difficulty in definitively addressing concerns regarding false positives generated by the Oncomine^™^ Focus Assay. Comparing the detection limits of these three methodologies, the Oncomine^™^ Focus Assay and Pan Lung Cancer PCR Panel utilize allele frequency thresholds of 5% and ˂1%, respectively, whereas the LDT RT-PCR method was developed by our team and lacks relevant data [10,15,16]. In addition, results from the LDT RT-PCR were interpreted by the naked eye, which may increase either bias or uncertainties. Nonetheless, based on these data, the LDT RT-PCR method was inferred to be the least sensitive of the three methods. Recent studies have suggested that the deletion of thymidine repeats at the *MET*ex14 donor site may affect the accuracy of the analysis, potentially leading to false positives [7]. This could explain the inconsistency in the results of the Oncomine^™^ Focus Assay and Pan Lung Cancer PCR Panel for samples with lower read counts.

One limitation of this study was that the current sample size did not allow for comprehensive verification of the false-positive occurrence rate of *MET*ex14 fusion. Therefore, it is necessary to establish a proper *MET*ex14 skipping quality control process. Based on our retrospective analysis, it is recommended to first confirm sequence integrity using IGV, especially for samples with lower read counts. If the ratio of read counts to total fusion reads is ≤0.25%, validation should be conducted using RT-PCR to enhance the reliability of the results.

## 5. Conclusions

The results of this study indicate that specimens with a low Oncomine^™^ Focus Assay read count to total fusion read ratios should be considered false-positive cases. Based on our findings, it is suggested that the Oncomine^™^ Focus Assay read counts and ratios should be reviewed, and for low ratios, the results should be validated using RT-PCR to prevent false positives. This can help accurately diagnose *MET*ex14 skipping and treat NSCLC effectively.

## Figures and Tables

**Figure 1 diagnostics-14-01110-f001:**
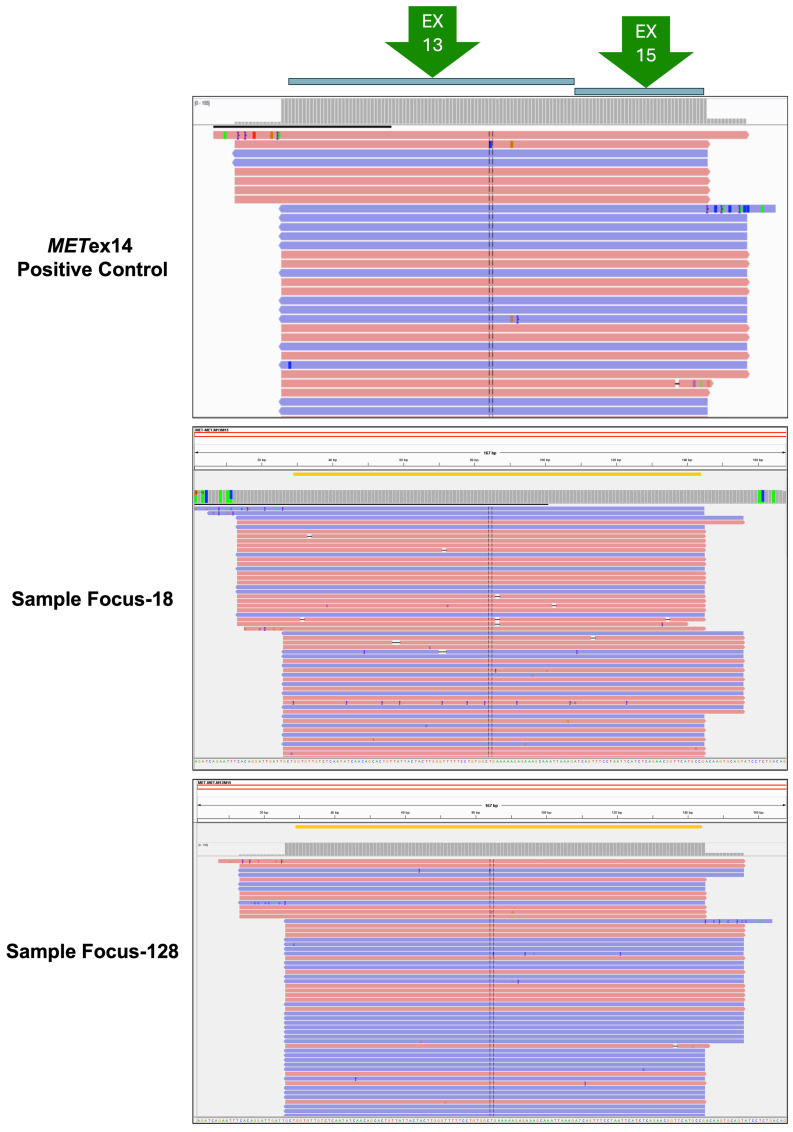
Visualization of two samples from Oncomine^™^ Focus Assay-positive cases using the Integrative Genomics Viewer. Two samples selected from 11 Oncomine^™^ Focus Assay-positive cases and 1 *MET*ex14 positive control were visualized using IGV after alignment with the human genome. Each sample contained multiple reads, all of which showed that the *MET* exons 13 and 15 were fused completely without exon 14. Symbols including each base or dash inside in each reads represent the SNP locations. The reads color and direction represent different strands; pink represent sense strand and purple represent anti-sense strand.

**Figure 2 diagnostics-14-01110-f002:**
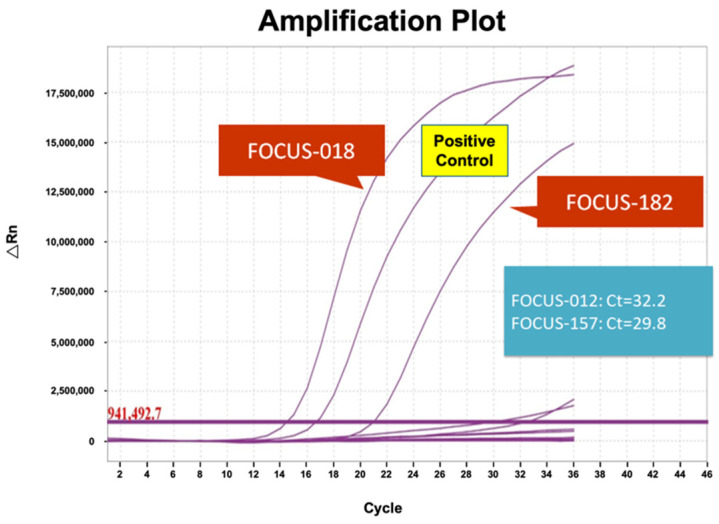
Amplification plots of 11 Oncomine^™^ Focus Assay-positive specimens. Pan Lung Cancer PCR Panel amplification plot of 11 Oncomine^™^ Focus Assay-positive specimens. Two samples, FOCUS-018 and FOCUS-182, showed cycle threshold (Ct) values below 28, similar to the positive control sample. The remaining nine samples had Ct values > 28 and were considered negative for the *MET*ex14 skipping mutation-containing specimens.

**Figure 3 diagnostics-14-01110-f003:**
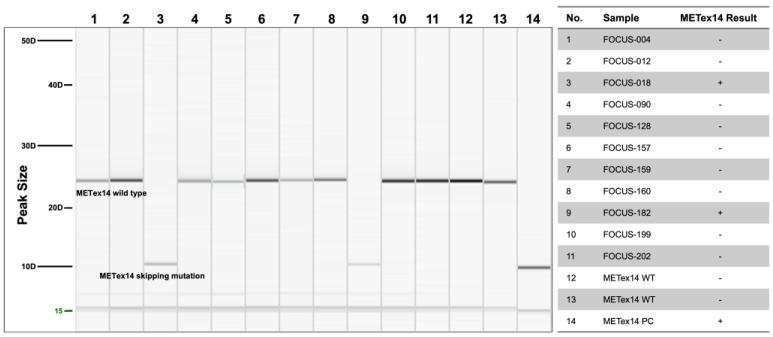
Electrophoresis of the 11 Oncomine^™^ Focus Assay-positive specimens. Electrophoresis of the 14 samples, including 11 Oncomine^™^ Focus Assay-positive and 3 control specimens. Lanes 3 and 9 showed molecular weights identical to that of lane 14, indicating that both samples contained the *MET*ex14 skipping mutation. The other nine samples showed the same molecular weight as the two wild-type control specimens (lanes 12 and 13), indicating that their *MET* genes were identical to that of the WT.

**Table 1 diagnostics-14-01110-t001:** The three platforms used for *MET*ex14 skipping mutation detection.

Sample	Oncomine Focus Assay			AmoyDx (Ct < 28)	RT-PCR
	Read Count (Cut Off >120)	Total Mapped Fusion Reads	Fusion Reads Count/Total Mapped Fusion Reads		
FOCUS-004	302	221,438	0.13%	-	-
FOCUS-012	612	280,337	0.22%	-	-
FOCUS-018	10,177	218,892	4.65%	+ *	+ *
FOCUS-090	261	318,617	0.08%	-	-
FOCUS-128	154	282,856	0.05%	-	-
FOCUS-157	201	153,230	0.13%	-	-
FOCUS-159	179	476,881	0.04%	-	-
FOCUS-160	319	581,357	0.05%	-	-
FOCUS-182	2540	197,851	1.28%	+ *	+ *
FOCUS-199	333	133,050	0.25%	-	-
FOCUS-202	212	122,505	0.17%	-	-

* The concordance rate of two positive cases on all three platforms was 18.2%. The symbol “+” and “-” represent positive and negative results.

## Data Availability

Data are contained within the article.

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
