# Peer review of "Retrospective Analysis to Optimize the Detection of *MET* Exon 14 Skipping Mutations in Non-Small Cell Lung Cancer"

_diagnostics, 2024, doi:10.3390/diagnostics14111110_

Round 1
Reviewer 1 Report
Comments and Suggestions for Authors
In this manuscript, the author evaluated positive cases tested using OFA and compared the results with those by the Pan Lung Cancer PCR Panel and reverse transcription (RT)-PCR on their own, aiming to optimize the routine testing for METex14. Here are some questions:
1.The author just validated and compared the LDT RT-PCR analyses with Oncomine™ Focus Assay results. It may be more convincing if they also compared the results with FoundationOne™ and Archer-MET™ which were already approved.
2.Line 157-158, in the second paragraph of discussion, why the author inferred the LDT RT-PCR was the least sensitive method?
3.The figures are not clear enough, please change them into a clear version.
4.The ethical declaration has not been included in this manuscript.
Author Response
1. The author just validated and compared the LDT RT-PCR analyses with Oncomine™ Focus Assay results. It may be more convincing if they also compared the results with FoundationOne™ and Archer-MET™ which were already approved.
Response:
We are appreciated for the suggestion. Due to the cost prices and policy, currently our hospital has not used FoundationOne™ or Archer-METTM as the lung cancer’s companion diagnostics kit, and we will consider this comparison in the future.
2. Line 157-158, in the second paragraph of discussion, why the author inferred the LDT RT-PCR was the least sensitive method?
Response:
Based on two reasons, we consider that LDT RT-PCR was the leas sensitive method. First, LDT RT-PCR without of the allele sensitivities of frequency thresholds, which consequence to the evaluation difficulty. Secondary, results of LDT RT-PCR were interpretated by naked eyes which may increase either bias or uncertainties. The second reason that we have rephrased and added in the discussion section in Lines 172-173.
3.The figures are not clear enough, please change them into a clear version.
Response:
We have replaced the figures and changed into much higher resolution images.
4. The ethical declaration has not been included in this manuscript.
Response:
Due to our analysis was a retrospective cohort study with anonymized paraffin-embedded tissues, which has no necessary to apply ethical declaration.
Reviewer 2 Report
Comments and Suggestions for Authors
Current data shows that MET proto-oncogene, receptor tyrosine kinase (MET) exon 14 skipping mutations is present in up to 4% of advanced NSCLC. This genetic anomaly can be addressed by novel tyrosine kinase inhibitors.
The current paper discusses optimized MET ex14 skipping mutation detection by analyzing 223 Oncomine™ 10 Focus Assay-positive cases using Pan Lung Cancer PCR Panel and reverse transcription (RT)-PCR.
I suggest the authors to develop in the introduction section on next generation sequencing- current data, on the differences between Illumina and Oncomine platorms. Also, to add data on the clinical importance of this mutation and what are the current drugs addressing this kind of mutations.
The materials and method sections needs reviewing.
Please mention if the patients’ data were anonymized.
Please develop on the protocol used- blood samples collection, the panel of genes that were tested for all patients.
Please add the inclusion and exclusion criteria, the number and percentage of MET exon 14 skipping mutation patients.
Please add the Ethics Committee Approval Number for the study.
Author Response
I suggest the authors to develop in the introduction section on next generation sequencing- current data, on the differences between Illumina and Oncomine platforms. Also, to add data on the clinical importance of this mutation and what are the current drugs addressing this kind of mutations.
Response:
We have added few descriptions about the differences between these two platforms in Lines 57-61. We also added the importance of this comparison data in Lines 67-68, which can be avoided unnecessary medication on those false-positive case patients.
The materials and method sections need reviewing.
Response:
We have edited this section and added descriptions about the rationale of this study in Lines 78-82, and Lines 84-86. The term “patient’s tissue” in Line 91 was replaced to “paraffin-embedded tissues”.
Please mention if the patients’ data were anonymized.
Response:
We have added the description to mention that all study materials were anonymized in Lines 78-82.
Please develop on the protocol used- blood samples collection, the panel of genes that were tested for all patients.
Response:
We have added the description about the materials in this retrospective cohort study in Lines 78-79, which were paraffin-embedded tissues.
Please add the inclusion and exclusion criteria, the number and percentage of MET exon 14 skipping mutation patients.
Response:
We have added the criteria description in Lines 78-82, which were wrote as following sentences: This retrospective cohort study analyzes previously stored anonymized paraffin-embedded tissues from NSCLC positive cases at Chang Gung Memorial Hospital, spanning 2019 to 2021. Tissue blocks that meet any of the following criteria will be excluded: 1. tumor cellularity (tumor content) below 30%, 2. tumor area less than 125 mm², 3. fewer than five blocks or blocks without a 10-microliter thickness.
Please add the Ethics Committee Approval Number for the study.
Response:
Due to our analysis was a retrospective cohort study with anonymized paraffin-embedded tissues, which has no necessary to apply ethical declaration.
Round 2
Reviewer 1 Report
Comments and Suggestions for Authors
The ethical declaration has not been included in this manuscript.
Author Response
The ethical declaration has not been included in this manuscript.
Response:
Since our analysis was a retrospective cohort study with anonymized paraffin-embedded tissues, we have not been asked to apply ethical declaration and we don’t have this declaration.
Reviewer 2 Report
Comments and Suggestions for Authors
The article can be accepted in the current form
Author Response
The article can be accepted in the current form
Response:
We appreciate reviewer's comment and feedback.